# Introduction of a Post-Anaesthesia Care Unit in a Teaching Hospital Is Associated with a Reduced Length of Hospital Stay in Noncardiac Surgery: A Single-Centre Interrupted Time Series Analysis

**DOI:** 10.3390/jcm13020534

**Published:** 2024-01-17

**Authors:** Nick J. Koning, Joost L. C. Lokin, Lian Roovers, Jan Willem Kallewaard, Wim H. van Harten, Cor J. Kalkman, Benedikt Preckel

**Affiliations:** 1Department of Anesthesiology and Pain Medicine, Rijnstate Hospital, Wagnerlaan 55, 6815 AD Arnhem, The Netherlands; 2Department of Anesthesiology, Radboud University Medical Centre, 6525 GA Nijmegen, The Netherlands; 3Clinical Research Center, Rijnstate Hospital, 6815 AD Arnhem, The Netherlandsw.h.vanharten@utwente.nl (W.H.v.H.); 4Department of Anesthesiology, Amsterdam University Medical Centre, 1105 AZ Amsterdam, The Netherlands; 5Health Services & Technology Research, University of Twente, 7522 NB Enschede, The Netherlands; 6Department of Anesthesiology, University Medical Centre Utrecht, 3584 CX Utrecht, The Netherlands

**Keywords:** anaesthesia recovery period, intensive care units, post-anaesthesia care unit, postoperative care, postoperative complications, surgical procedures, operative

## Abstract

Background: A post-anaesthesia care unit (PACU) may improve postoperative care compared with intermediate care units (IMCU) due to its dedication to operative care and an individualized duration of postoperative stay. The effects of transition from IMCU to PACU for postoperative care following intermediate to high-risk noncardiac surgery on length of hospital stay, intensive care unit (ICU) utilization, and postoperative complications were investigated. Methods: This single-centre interrupted time series analysis included patients undergoing eleven different noncardiac surgical procedures associated with frequent postoperative admissions to an IMCU or PACU between January 2018 and March 2019 (IMCU episode) and between October 2019 and December 2020 (PACU episode). Primary outcome was hospital length of stay, secondary outcomes included postoperative complications and ICU admissions. Results: In total, 3300 patients were included. The hospital length of stay was lower following PACU admission compared to IMCU admission (IMCU 7.2 days [4.2–12.0] vs. PACU 6.0 days [3.6–9.1]; *p* < 0.001). Segmented regression analysis demonstrated that the introduction of the PACU was associated with a decrease in hospital length of stay (GMR 0.77 [95% CI 0.66–0.91]; *p* = 0.002). No differences between episodes were detected in the number of postoperative complications or postoperative ICU admissions. Conclusions: The introduction of a PACU for postoperative care of patients undergoing intermediate to high-risk noncardiac surgery was associated with a reduction in the length of stay at the hospital, without increasing postoperative complications.

## 1. Introduction

Over the last decades, the intraoperative period has become very safe even for high-risk surgical patients [1,2,3]. In the postoperative period, however, rates for adverse events resulting in potentially preventable morbidity and mortality remain high [4,5,6]. Postoperatively, only a minority of high-risk surgical patients is routinely admitted to critical care facilities [7]. Close monitoring of high-risk patients in the early postoperative period is pivotal to prevent and detect patient deterioration and complications in an early stage and thus avert detrimental outcomes. For example, myocardial injury may be preceded by a modifiable risk factor such as postoperative hypotension which could be detected by trained nursing staff, and supervision by skilled nursing staff can decrease postoperative respiratory complication rates in patients with obstructive sleep apnoea [8,9,10]. Many hospitals routinely admit high-risk surgical patients to the intensive care unit (ICU) or intermediate care unit (IMCU) for continuous monitoring and timely treatment of postoperative complications. Unlike critical care, the level of care at an IMCU is not well characterized, and large variations can exist between different wards all designated as an IMCU. Nonetheless, the introduction of an IMCU has been shown to reduce ICU occupancy without affecting the quality of postoperative care [11].

A post-anaesthesia care unit (PACU) is a facility with advanced monitoring where high-risk surgical patients receive specialized perioperative care. They have received increasing attention as a measure to reduce pressure on scarce ICU capacity [12,13]. Both IMCUs and ICUs often receive a mix of medical and surgical patients, whereas a PACU is entirely dedicated to postoperative care. Moreover, the duration of stay in the PACU is typically tailored to the individual patients’ needs, which facilitates enhanced recovery after surgery [14]. In contrast, in most IMCUs and ICUs, patients routinely stay at least one night.

In the current study, we report the effects of a hospital-wide transition from IMCU to PACU for postoperative care in patients undergoing intermediate- or high-risk noncardiac surgery on hospital length of stay, unanticipated postoperative ICU admission, and postoperative complications. In order to account for potential time bias and confounding by case mix differences, segmented regression analysis was performed in an interrupted time series design [15]. We hypothesized that postoperative care in a PACU is associated with a reduction in the length of stay at the hospital as compared to a mixed medical-surgical IMCU, without increasing postoperative ICU admissions or postoperative complication rate.

## 2. Materials and Methods

### 2.1. Study Design

The current single-centre retrospective interrupted time series analysis was conducted in a Dutch teaching hospital (Rijnstate Hospital, Arnhem, The Netherlands). The medical research ethics committee Oost-Nederland (Chairperson: prof. dr. P.N.R. Dekhuijzen) reviewed the current study protocol (CCMO register number: 2021-13068) and waived the requirement for ethical approval due to the observational nature of the study on 9 August 2021. Registered objection by the patient to use registered data for medical research was checked in the Electronic Health Record (EHR). The need for written informed consent was waived after absence of objection, according to hospital protocol. This manuscript was prepared in accordance with the Strengthening the Reporting of Observational Studies in Epidemiology (STROBE) statement [16].

### 2.2. Patient Selection

Eleven different surgical interventions for which postoperative patients were frequently admitted to IMCU were identified using procedural codes:(1)Thyroidectomy(2)Parathyroidectomy(3)Video-Assisted Thoracoscopic Surgery (VATS) for (bi)lobectomy(4)Open (bi)lobectomy(5)Endovascular aortic repair (EVAR) or Covered Endovascular Reconstruction of Aortic Bifurcation (CERAB)(6)Laparoscopic colonic or rectal resection(7)Cystectomy with urinary diversion(8)Debulking for endometrial or ovarian carcinoma(9)Hip fracture surgery(10)Revision of total hip or total knee arthroplasty(11)Arterial revascularization or embolectomy of the lower limb.

All adult patients who underwent one of these procedures between the 1 January 2018 and the 31 March 2019 (ICMU episode) and between the 1 October 2019 and 31 December 2020 (PACU episode) were included in this study for analysis, independently of whether patients were indeed admitted to an IMCU or PACU postoperatively.

Only the first surgical procedure for each patient was included; subsequent surgeries for included patients were excluded from the current analysis.

### 2.3. Preoperative Care

All elective patients were screened and informed about the scheduled operation on the preoperative outpatient clinic. Perioperative risk assessment and additional investigations or consultations were performed based on current guidelines [17,18]. At the preoperative outpatient clinic, the anaesthesiologist indicated the appropriate postoperative destination for the patient based on a combination of the type of surgery and patient risk factors. Only patients undergoing pulmonary surgery were always scheduled for admission to the IMCU, the PACU, or, rarely, the ICU, irrespective of patient risk factors.

### 2.4. Postoperative Care Setting

Selection for admission to the either the IMCU or the PACU was left to the discretion of the anaesthesiologist at the preoperative screening clinic in combination with the estimation of the attending anaesthesiologist and the perioperative team including the surgeon on the day of surgery in case of unforeseen events.

The characteristics of the IMCU and PACU departments are presented in Table 1.

Before implementation of the PACU, high-risk surgical patients were admitted to the IMCU—or alternatively to the ICU—for postoperative care. Planned postoperative ICU admission was requested only for patients with the highest risk, including an anticipated need for postoperative mechanical ventilation. The IMCU had nine beds for postoperative patients on weekdays and three beds during the weekend (day and night), in addition to seven beds for medical patients. Before transfer to IMCU or ICU, postoperative patients first stayed at the recovery room (for at least 30 min) under supervision of the attending anaesthesiologist until they were deemed sufficiently stable for further transport.

Continued postoperative care at the IMCU was managed by specialized nurses with a nurse-to-patient ratio of 1:3 to 1:4 and a surgeon or surgical resident was responsible for postoperative care. Arterial and central venous lines that were placed intraoperatively could be used for invasive cardiovascular monitoring and support with vasopressors and/or inotropes. Neither mechanical ventilation nor non-invasive ventilation was available at the IMCU. In case of serious adverse events or a medical emergency, the rapid response team from the ICU was available for consulting. If (non)-invasive ventilation was indicated, the patient was transferred to the ICU. Patients stayed overnight and were discharged the first postoperative morning to the surgical ward at the discretion of the supervising surgeon or surgical resident.

The PACU was introduced in July 2019—first for a 3-month transition period—which was completed in September 2019. In contrast to the IMCU, this unit is situated on the operation rooms floor and is located directly adjacent to the recovery room. The PACU has eight beds during weekdays (day and night) and is closed on the weekends from Saturday afternoon until Monday morning. Nurses previously working in the recovery room stayed as PACU nursing staff and were given additional training in prolonged postoperative care before the transition to the PACU as needed. All high-risk patients with an estimated uncomplicated recovery within 24 h postoperatively were planned for PACU admission. Only if the need for prolonged critical care was anticipated (at the discretion of the treating anaesthesiologist), the patients were admitted to the ICU directly after surgery. Haemodynamic monitoring and support and/or (non-) invasive mechanical ventilation could be provided in the PACU. PACU nursing staff delivered both regular recovery room care and specialized postoperative care to high-risk patients on the PACU. The nurse-to-patient ratio varied between 1:2 to 1:4; it was higher during the day and lower at night. During daytime, the anaesthesiologist who treated the patient intraoperatively remained responsible for postoperative care at the PACU, in close collaboration with the surgical staff. After working hours, patients on the PACU were supervised by the on-call anaesthesiologist, supported by a surgical resident. The anaesthesiologist is physically always close to the PACU. In the Dutch healthcare system, anaesthetic nurses stay in the OR during surgery and the anaesthesiologist is therefore rarely occupied on the OR without possibilities to leave the OR for an extended continuous timeframe.

The required length of stay at the PACU was determined by the (treating) anaesthesiologist. If, 24 h after surgery, the patient still required a higher level of care than can be provided at the regular surgical ward, ICU staff was contacted to discuss prolonged PACU stay or ICU admission for further treatment.

### 2.5. Data Collection

Patient selection was performed retrospectively from the hospital EHR based on Dutch Healthcare Authority procedural codes (NZa activity codes) for the relevant surgical procedures with the timeframes 1 January 2018 to 31 March 2019 (IMCU episode) and 1 October 2019 to 31 December 2020 (PACU episode). For the optimal comparison of care at both postoperative care settings, the six-month transition period (April 2019–September 2019) was excluded from data analysis.

Patient characteristics, hospital admission data, and details of surgical procedures were extracted from the EHR. Hospital admission data included length of hospital stay, the department to which the patient was admitted, and the duration of stay for each department. Extraction and definitions of patients’ pre-existing conditions and postoperative complications is described in Appendix A. In brief, pre-existing conditions were obtained using registered diagnoses. In addition, relevant preoperative laboratory data and current medication were obtained to identify pre-existing conditions. Postoperative complications occurring within 30 days postoperatively were identified using new postoperative diagnoses, relevant activity codes registered in the EHR, or using relevant postoperative laboratory data. All postoperative ICU admissions were reviewed for identification of postoperative complications and for indication of ICU admission. In particular, ICU admissions were divided in admission with or without a strictly medical ICU indication (e.g., due to insufficient capacity at the IMCU or PACU).

### 2.6. Outcomes

The primary outcome of this study was the hospital length of stay. Secondary outcomes included (1) duration of stay at the PACU or IMCU, (2) the percentage of patients admitted to the PACU or IMCU, (3) rate of postoperative ICU admissions, (4) unplanned reoperations within 30 days after the initial surgical procedure, (5) hospital readmission within 30 days after surgery, (6) mortality at 30 days, 3 months, and 1 year after surgery, and (7) postoperative complications within 30 days after surgery (Appendix A). All outcomes were compared between the two different episodes (IMCU vs. PACU) per surgical intervention group.

### 2.7. Statistical Analysis

Statistical analysis was performed with IBM SPSS Statistics (version 22, 2013, IBM, New York City, NY, USA) and with R 4.2.3 (R Foundation, Vienna, Austria). Continuous data were tested for normal distribution with Kolmogorov–Smirnov test and presented as means and SD or median [IQR] when parameters were not normally distributed. Two-sided Mann–Whitney U test was used to evaluate differences in continuous data between groups, whereas Fisher’s exact test was used for nominal data. Post hoc correction was used when applicable.

For the primary analysis, a segmented regression model was fitted to account for preintervention trends and confounding variables [15]. The model included study period (pre- vs. post-implementation of the PACU), admittance to specialized (IMCU/PACU) vs. regular ward postoperatively, period by specialized or regular ward interaction, time, age, gender, BMI, number of medications, ASA-PS class, pre-existing conditions, type of surgery, and surgical duration. Natural logarithmic transformation was performed on hospital length of stay. Age, BMI, number of medications, and surgical duration were centred on the overall median value, whereas ASA-PS class and surgical type were centred on the most prevalent value. To back-transform results onto the untransformed scale of measurement, we exponentiated regression coefficients and reported them as the geometric mean ratios (GMR) with 95% confidence intervals (95% Cis). Patients with missing data were excluded for the segmented regression analysis. In a sensitivity analysis, the patients who died during admission or who were discharged to another hospital, or a hospice were excluded. Additionally, a sensitivity analysis was performed with postoperative length of stay as outcome measure instead of total hospital length of stay. A *p*-value < 0.05 was considered statistically significant.

## 3. Results

### 3.1. Patient Characteristics

In total, 3300 patients were included in the current study. During the IMCU episode, 1746 patients underwent one of the eleven pre-defined high-risk surgical procedures and were thus included for analysis. Of these patients, 678 (38.8%) were actually admitted to the IMCU. During the PACU episode, 1554 patients were included, of which 625 (40.2%; *p* = 0.433 between groups) were admitted to the PACU.

Overall, patient characteristics were similar between episodes (Table 2). Revascularizations of the lower extremity were more frequently encountered in the IMCU cohort, whereas hip fracture repair surgeries were more prevalent in the PACU cohort.

### 3.2. Hospital Length of Stay

Hospital length of stay was longer in patients following IMCU admission when compared to patients who were admitted to the PACU (7.2 days [4.2–12.0] vs. 6.0 days [3.6–9.1], respectively; *p* < 0.001). A reduction in hospital length of stay for the subset of patients admitted to the PACU compared to the subset of patients admitted to the IMCU was observed for eight out of eleven different surgical procedures (Figure 1).

Total hospital length of stay for all included patients was slightly shorter during the PACU episode compared to the IMCU episode (IMCU 5.6 days [3.3–9.3] versus PACU 5.2 days [3.1–8.2], *p* < 0.001).

### 3.3. Segmented Regression Analysis

The linear segmented regression predicting the relationship between hospital length of stay and relevant covariates is shown in Table 3. Segmented regression analysis was performed on 3115 patients, after excluding 185 patients with missing BMI values for this analysis.

Hospital length of stay did not change over time in the IMCU period (confounder adjusted geometric mean ratio [GMR] of 1.00 per month [95% CI 1.00–1.01]; *p* = 0.374). In the PACU episode, hospital length of stay slightly decreased over time (GMR 0.99 per month [95% CI 0.98–1.00]; *p* = 0.034). For patients admitted to a specialized ward (IMCU or PACU) postoperatively, the transition from the IMCU to PACU episode was associated with a decrease in hospital length of stay by 23% (GMR 0.77 [95% CI 0.66–0.91]; *p* = 0.002).

Other parameters associated with decreased hospital length of stay were lower ASA-PS class, higher BMI, and multiple types of surgery. Conversely, pre-existing heart failure, atrial fibrillation, insulin-dependent diabetes, and anaemia were predictors associated with increased length of stay. The sensitivity analyses excluding patients who died during admission or who were discharged to another hospital or a hospice and using postoperative length of stay as outcome measure produced similar results compared to the primary analysis.

In Figure 2, the time effect during the study period on hospital length of stay is shown for patients during the IMCU and PACU episodes, with or without admission to a specialized ward postoperatively. For patients not admitted to an IMCU or PACU postoperatively, there was no difference in length of stay between both episodes. Patients admitted to a specialized ward postoperatively had a shorter length of hospital stay after transition from the IMCU to PACU. Additionally, a decrease in the slope (decreasing time trend) was observed only after introduction of the PACU.

### 3.4. Duration of Stay in the IMCU or PACU

There was no difference in the overall percentage of high-risk patients admitted to either IMCU or PACU (Table 4). Patients undergoing parathyroidectomy and hip fracture repair surgery were admitted less often to the PACU than to the IMCU, whereas more patients undergoing endovascular aortic aneurysm repair, cystectomy, and debulking for gynaecologic malignancy stayed in the PACU more frequently.

The duration of stay in the PACU was shorter and showed more variability than IMCU stay duration (IMCU: 20 h (17–23) vs. PACU: 6 h (4–19), *p* < 0.001; coefficient of dispersion 0.15 vs. 0.65), as shown in Figure 3.

Admission to the IMCU was always preceded by direct postoperative care in the recovery room for almost 2 h (111 min (87–144)), whereas PACU admission occurred directly after surgery. Similarly, fewer patients stayed overnight at the PACU (275 patients; 44% of admissions) compared to the IMCU (659 patients; 97% of IMCU admissions; *p* < 0.001).

### 3.5. ICU Admissions and Postoperative Complications

The rate of total ICU admissions did not differ between episodes (IMCU episode 4.0% vs. PACU episode 4.1%; *p* = 0.929), nor did the number of ICU admissions with a strict medical ICU indication (IMCU episode 3.8% vs. PACU episode 2.6%; *p* = 0.062; Table 5). However, following IMCU or PACU admission more patients were subsequently admitted to the ICU with a strictly medical indication (IMCU episode 7.4% vs. PACU episode 4.6%; *p* = 0.048; Appendix A).

The incidence of postoperative complications was similar during the IMCU and PACU episodes (8.5% vs. 9.6%, respectively; *p* = 0.273). New-onset atrial fibrillation was more frequently detected during the PACU episode (IMCU 1.9% vs. PACU 3.1%; *p* = 0.043), whereas hospital readmission rate was higher during the IMCU episode (IMCU 14.7% vs. PACU 12.1%; *p* = 0.032). No differences between episodes in other postoperative complications were observed.

For the subset of patients admitted to either the IMCU or the PACU, increased mortality at 3 months (IMCU 9.1% vs. PACU 6.1%; *p* = 0.047) and 1 year after surgery (IMCU 16.2% vs. PACU 11.8%; *p* = 0.026) and acute kidney injury incidence (IMCU 4.4% vs. PACU 2.4% *p* = 0.049) was observed following IMCU admission than after PACU admission (Appendix A). The incidence of all other postoperative complications was similar between patients admitted to the PACU or IMCU.

## 4. Discussion

We studied the effects of a transition from a mixed surgical/medical IMCU to PACU for postoperative care of intermediate to high-risk noncardiac surgical patients on hospital length of stay using an interrupted time series analysis. Segmented regression analysis demonstrated a decrease in the hospital length of stay following admission to the PACU as compared to admission to the IMCU by 23%. The reduction in the hospital length of stay following the introduction of the PACU was independent of case mix differences. No preintervention trend as a decrease in the hospital length of stay during IMCU episode was observed. Patients typically stayed overnight at the IMCU, whereas length of stay at the PACU was more individualized, which led to 50% of patients being discharged to the ward on the day of surgery. This shorter duration of stay at a high-care facility combined with a reduction in overall hospital length of stay was not accompanied by an increased rate of unplanned ICU admissions, reoperations, or postoperative complications, with the exception of new-onset atrial fibrillation, which was detected more often in the PACU episode. However, when focusing specifically on the subset of patients who were indeed admitted to the PACU, a lower incidence in acute kidney injury and mortality at 3 months and 1 year postoperatively was observed as compared to patients admitted to the IMCU.

The definition of a PACU is not uniform, which hinders comparison between different studies. For example, the PACU described by Thenuwara et al. may be seen as a regular recovery room for direct postoperative care [19], whereas the PACU as described in other studies provides prolonged specialized postoperative care—including possible mechanical ventilation—and is similar to the PACU in the current study [12,13,14,20]. A recent systematic review investigating the organization of postoperative care on patient outcome concluded that postoperative care at the PACU was not associated with worse outcomes compared to ICU care [21]. Several studies in cardiac surgery have focused on “fast-track recovery” on the PACU instead of conventional postoperative care on an ICU [14,20]. In these studies, postoperative care at the PACU led to shorter times until postoperative tracheal intubation and reduced the hospital length of stay when compared to direct ICU admission, while maintaining patient safety and high quality of care [14,20]. Similar to “fast-track recovery” in cardiac surgery, individualized postoperative care on the PACU for noncardiac surgery patients focusses on adequate pain management, early detection of patient deterioration, and treatment of any postoperative complications, which facilitates early mobilization and is delivered by nursing staff familiar with ERAS principles [12]. Moreover, the PACU is usually situated directly adjacent to the operating room (OR) complex, which facilitates continuity of care by the anaesthesiologist and multidisciplinary treatment by both the anaesthesiologist and surgeon.

The implementation of both IMCU and PACU care models resulted from a need to provide postoperative patient care on a level between that of ward care and intensive care. In addition, such intermediate care levels are one way of relieving pressure on scarce and costly ICU capacity. In the United Kingdom, more than 80% of high-risk surgical patients are not admitted to an ICU postoperatively [7]. Although it is unknown whether this is related to a shortage of ICU beds, it indicates the potential added value of having sufficient capacity for postoperative care on specialized wards outside an ICU. Indeed, recent studies from Australia indicate that an advanced recovery room care similar to the PACU in the current study appears to increase days alive at home after medium risk noncardiac surgery in a cost-effective manner when compared to postoperative care in a regular ward [22,23].

Importantly, our data indicate that the shorter duration of stay at the PACU—as compared to the longer stay at the IMCU—was not accompanied by an increase in postoperative complications. Similarly, a previous study showed that the introduction of a standardized clinical pathway for patients at a regular recovery roomled to both improved outcomes for all noncardiac surgical patients and a reduction in length of stay following uneventful surgery [24]. The present study found that anaesthesiologists considered more than half of patients admitted to the PACU ready for discharge to the ward on the day of surgery. This may have contributed to the decrease in total hospital length of stay after introduction of the PACU. Moreover, the early discharge of patients allows for either less scheduled nursing staff at nighttime or leave capacity for patients needing PACU admission after undergoing emergency surgery at night.

Interestingly, for several surgical indications, including cystectomy with urinary diversion, debulking for gynaecologic malignancies, and endovascular aneurysm repair, an increase in postoperative admissions to the PACU was observed after the transition from the IMCU to the PACU. Altogether, this individualization in the duration of monitored postoperative care resulted in a shorter median length of stay in the hospital and did not increase the rate of postoperative adverse events. Although new-onset atrial fibrillation was detected more frequently in the PACU episode, this was no longer observed when selecting only patients admitted to the PACU or IMCU. In contrast, admission to the PACU was associated with lower rates of acute kidney injury and mortality at 3 months and 1 year as compared to admission to the IMCU.

### Limitations

Our study has several limitations. First, the retrospective design of the study is a potential source of bias. However, segmented regression analysis in our interrupted time series analysis deemed relevant preintervention trends or case mix differences between episodes unlikely to explain the decrease in the hospital length of stay following the transition from IMCU to PACU.

Second, this was a single centre study and whether our results are generalizable to other medical centres, with different sizes and case mixes or with different perioperative care logistics, is unknown. Since worldwide differences in organization of postoperative care are considerable, the effects of the introduction of a PACU may differ per country. However, the current before–after study demonstrates that a PACU may be a high-quality and efficient solution for hospitals with a limited number of ICU or IMCU beds. Tailoring the duration of stay at the PACU to the individual patients’ needs allows for delivering intensive postoperative care to an increasing number of high-risk surgical patients with a relatively limited increase in nursing staff and resources. Although we did not assess cost-effectiveness in the current study, an economic analysis of an introduction of a PACU would be an interesting topic for further study.

Finally, although in some countries, discharge practices may be influenced by a reimbursement policy, there was no change in economic consequences for the hospital over the research period in the current study. Since postoperative complications and readmissions did not increase after the introduction of the PACU, there was no indication of an inappropriate reduction in the hospital length of stay. Other healthcare reimbursement systems may, however, influence the generalizability of the current findings to these systems.

## 5. Conclusions

Transition from a mixed surgical/medical IMCU to a PACU for postoperative care of patients undergoing intermediate to high-risk noncardiac surgery was associated with a reduction in the hospital length of stay for a wide range of surgical interventions in the current interrupted time series analysis. Individualized postoperative duration of stay in the PACU resulted in a return to the regular ward on the day of surgery for half of the patients, without increasing risk for postoperative complications. The generalizability of the current results may, however, be limited due to the observational nature of this single-centre study.

## Figures and Tables

**Figure 1 jcm-13-00534-f001:**
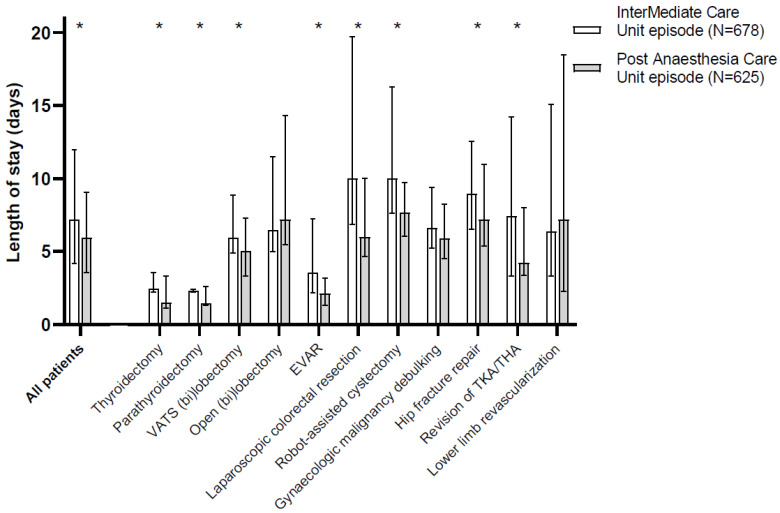
Length of stay. Hospital length of stay for patients admitted to an intermediate care unit (IMCU; white bars) or post-anaesthesia care unit (PACU; grey bars) postoperatively. Only patients who were admitted to the IMCU (*n* = 678) or PACU (*n* = 625) are included in the current data. Data are presented as median with interquartile range. CERAB = Covered Endovascular Reconstruction of Aortic Bifurcation, EVAR = Endovascular aortic aneurysm repair, THA = total hip arthroplasty, TKA = total knee arthroplasty, VATS = Video-assisted thoracoscopic surgery. * *p* < 0.05 between episodes as tested with Mann–Whitney U tests.

**Figure 2 jcm-13-00534-f002:**
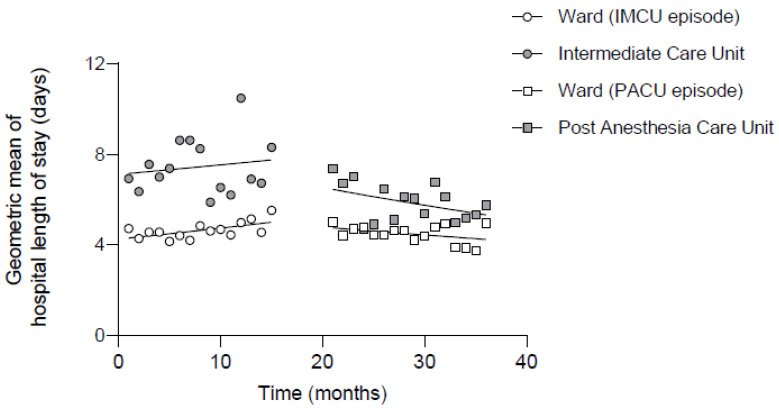
Hospital length of stay per month. The geometric mean of hospital length of stay per month is presented for patients with and without admission to an IMCU or PACU postoperatively during the intermediate care unit and post-anaesthesia care unit episodes. The trend lines were fitter using segmented regression analysis. IMCU = Intermediate Care Unit, PACU = Post Anaesthesia Care Unit.

**Figure 3 jcm-13-00534-f003:**
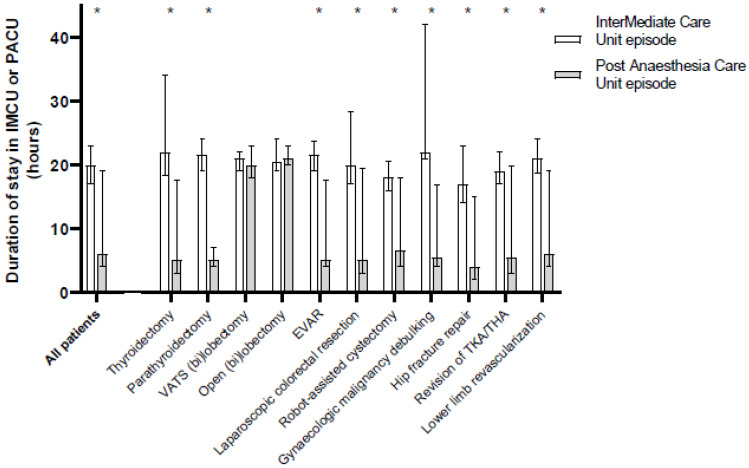
Duration of stay in intermediate care or post-anaesthesia care unit. Duration of stay in the intermediate care unit (IMCU; white) or post-anaesthesia care unit (PACU; grey) in hours following different surgical procedures. Only patients who were admitted to the IMCU (*n* = 678) or PACU (*n* = 625) are included in the current data. Data are presented as median with interquartile range. CERAB = Covered Endovascular Reconstruction of Aortic Bifurcation, EVAR = Endovascular aortic aneurysm repair, THA = total hip arthroplasty, TKA = total knee arthroplasty, VATS = Video-assisted thoracoscopic surgery. * *p* < 0.05 between episodes as tested with Mann–Whitney U tests.

**Table 1 jcm-13-00534-t001:** Intermediate care unit and post-anaesthesia care unit characteristics.

	Intermediate Care Unit (IMCU)	Post-Anaesthesia Care Unit (PACU)
Nurse to patient ratio	1:3 to 1:4	1:2 to 1:4
Beds available	9	8
Opening hours	7 days per week, 24 h per day	Monday morning to saturday afternoon, 24 h per day
Admission	After a recovery period in the recovery room	Direct postoperatively
Medical supervision	Surgical staff	Anesthesiologist in collaboration with surgical staff
Catecholamines	Yes	Yes
(Non-)invasive ventilation or high-flow nasal oxygen	No	Yes
Duration of stay	Overnight	Individualized, based on the estimation of the anaesthesiologist

**Table 2 jcm-13-00534-t002:** Patient characteristics.

	Intermediate Care Unit Episode (*n* = 1746)	Post Anaesthesia Care Unit Episode (*n* = 1554)
Age	71 (61–80)	72 (62–81)
Gender (male/female)	44%/56%	42%/58%
ASA-PS class	3 (2–3)	3 (2–3)
BMI	25 (23–29)	25 (23–29)
Number of different medications	4 (1–9)	5 (1–11)
Pre-existing conditions		
Coronary artery disease	212 (12%)	164 (11%)
Congestive heart failure	114 (7%)	108 (7%)
Valvular heart disease	83 (5%)	98 (6%)
Atrial fibrillation	147 (8%)	160 (10%)
Insulin-dependent diabetes mellitus	127 (7%)	80 (5%)
COPD	132 (8%)	97 (6%)
Respiratory insufficiency other than COPD	73 (4%)	79 (5%)
Chronic renal failure	187 (11%)	162 (10%)
Stroke	103 (6%)	117 (8%)
Anaemia	238 (14%)	234 (15%)
Type of surgery		
Thyroidectomy	110 (6%)	91 (6%)
Parathyroidectomy	33 (2%)	29 (2%)
VATS (bi)lobectomy	70 (4%)	73 (5%)
Open (bi)lobectomy	36 (2%)	34 (2%)
Endovascular aortic aneurysm repair	153 (9%)	115 (7%)
Laparoscopic colorectal resection	388 (22%)	295 (19%)
Cystectomy with urinary diversion	80 (5%)	85 (6%)
Debulking for ovarian of uterine malignancy	74 (4%)	76 (5%)
Hip fracture repair	523 (30%)	578 (37%)
Revision of total knee or total hip arthroplasty	133 (8%)	105 (7%)
Revascularization of the lower extremity	146 (8%)	73 (5%)
Surgical duration (minutes)	91 (54–151)	78 (50–142)

ASA-PS = American Society of Anaesthesiology—Physical Status, BMI = body mass index, COPD = Chronic Obstructive Pulmonary Disease, VATS = Video-assisted thoracoscopic surgery. Values are medians (interquartile range) or number of patients (percentage).

**Table 3 jcm-13-00534-t003:** Linear segmented regression analysis on hospital length of stay.

Parameter	Geometric Mean Ratio	95% Confidence Interval	*p*-Value
PACU episode (vs. IMCU episode)	0.97	0.86–1.10	0.618
Postoperative admission to monitored ward	1.51	1.41–1.62	<0.001
Interaction postoperative admission to monitored ward during PACU episode	0.80	0.73–0.88	<0.001
Time during IMCU episode (per additional month)	1.00	1.00–1.01	0.374
Time during PACU episode (per additional month)	0.99	0.98–1.00	0.034
Age (per additional year)	1.01	1.01–1.01	<0.001
Gender (male)	1.02	0.97–1.08	0.377
BMI (per additional kg/m^2^)	0.99	0.98–0.99	<0.001
Number of different medications (per additional medicament)	1.00	1.00–1.00	0.949
ASA-PS Class			
ASA-PS 1	0.79	0.70–0.90	<0.001
ASA-PS 2	0.92	0.87–0.98	0.006
ASA-PS 3	Reference	-	-
ASA-PS 4	1.15	1.05–1.26	0.003
ASA-PS 5	1.09	0.61–1.93	0.777
Pre-existing conditions			
Myocardial ischemia	0.98	0.90–1.06	0.573
Decompensated heart failure	1.14	1.03–1.27	0.013
Valvular heart disease	1.01	0.91–1.12	0.869
Atrial fibrillation	1.11	1.01–1.21	0.024
COPD	0.98	0.89–1.08	0.714
Respiratory insufficiency other than COPD	1.11	0.99–1.24	0.073
Chronic renal failure	1.02	0.94–1.11	0.642
Insulin-dependent diabetes mellitus	1.16	1.05–1.29	0.003
Anaemia	1.14	1.06–1.22	<0.001
Stroke	1.06	0.97–1.17	0.216
Surgery type			
Thyroidectomy	0.33	0.29–0.36	<0.001
Parathyroidectomy	0.36	0.30–0.43	<0.001
VATS (bi)lobectomy	0.79	0.70–0.90	<0.001
Open (bi)lobectomy	1.01	0.85–1.19	0.938
Endovascular aortic aneurysm repair	0.34	0.31–0.38	<0.001
Laparoscopic colorectal resection	0.96	0.89–1.04	0.351
Cystectomy with urinary diversion	0.85	0.74–0.98	0.023
Debulking for ovarian of uterine malignancy	0.81	0.71–0.91	0.001
Hip fracture repair	Reference	-	-
Revision of total knee or total hip arthroplasty	0.73	0.66–0.81	<0.001
Revascularization of the lower extremity	0.58	0.52–0.64	<0.001
Surgical duration (per additional minute)	1.00	1.00–1.00	<0.001

ASA-PS = American Society of Anaesthesiology—Physical Status, BMI = body mass index, COPD = Chronic Obstructive Pulmonary Disease, IMCU = Intermediate Care Unit, PACU = Post Anaesthesia Care Unit, VATS = Video-assisted thoracoscopic surgery.

**Table 4 jcm-13-00534-t004:** Percentage of admissions to intermediate care unit or post-anaesthesia care unit.

	Intermediate Care Unit Episode (*n* = 1746)	Post Anaesthesia Care Unit Episode (*n* = 1554)	*p*-Value
All surgical procedures	678 (39%)	625 (40%)	0.433
Thyroidectomy	36 (33%)	21 (23%)	0.158
Parathyroidectomy	30 (91%)	9 (31%)	<0.001
VATS (bi)lobectomy	67 (96%)	73 (100%)	0.115
Open (bi)lobectomy	34 (94%)	34 (100%)	0.493
Endovascular aortic aneurysm repair	76 (50%)	72 (63%)	0.047
Laparoscopic colorectal resection	86 (22%)	81 (28%)	0.127
Cystectomy with urinary diversion	30 (38%)	60 (71%)	<0.001
Debulking for ovarian of uterine malignancy	23 (31%)	40 (53%)	0.008
Hip fracture repair	179 (34%)	165 (29%)	0.044
Revision of total knee or total hip arthroplasty	39 (29%)	32 (31%)	0.887
Revascularization of the lower extremity	78 (53%)	38 (52%)	0.886

VATS = Video-assisted thoracoscopic surgery. Values are number of patients (percentage).

**Table 5 jcm-13-00534-t005:** Patient outcome.

	Intermediate Care Unit Episode (*n* = 1746)	Post-Anaesthesia Care Unit Episode (*n* = 1554)	*p*-Value
Any postoperative complication	148 (8.5%)	149 (9.6%)	0.273
Myocardial ischemia	23 (1.3%)	12 (0.7%)	0.172
Decompensated heart failure	33 (1.9%)	23 (1.4%)	0.419
New-onset atrial fibrillation	34 (1.9%)	48 (3.1%)	0.043
PPC	57 (3.3%)	48 (3.1%)	0.843
Stroke	9 (0.5%)	13 (0.8%)	0.289
AKI	48 (2.7%)	35 (2.3%)	0.375
Need for RRT	1 (0.1%)	1 (0.1%)	1.000
Pulmonary embolism	11 (0.6%)	13 (0.8%)	0.542
ICU admission	69 (4.0%)	63 (4.1%)	0.929
ICU admission for medical indication	67 (3.8%)	41 (2.6%)	0.062
Reoperation	119 (6.8%)	108 (6.9%)	0.891
Readmission	258 (14.7%)	188 (12.1%)	0.032
Mortality at 30 days	49 (2.8%)	42 (2.7%)	0.915
Mortality at 3 months	103 (5.9%)	80 (5.1%)	0.361
Mortality at 1 year	188 (10.8%)	138 (8.9%)	0.070

AKI = acute kidney injury, ICU = intensive care unit, PPC = postoperative pulmonary complications, RRT = renal replacement therapy.

## Data Availability

The data presented in this study are available on request from the corresponding author. The data are not publicly available due to privacy regulations.

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
