# Peer review of "Introduction of a Post-Anaesthesia Care Unit in a Teaching Hospital Is Associated with a Reduced Length of Hospital Stay in Noncardiac Surgery: A Single-Centre Interrupted Time Series Analysis"

_jcm, 2024, doi:10.3390/jcm13020534_

Round 1
Reviewer 1 Report
Comments and Suggestions for Authors
Thank you very much for considering me as a potential reviewer. Optimal postoperative treatment of non-cardiac surgical patients in times of reduced ICU resources are of high interest. Thus, this manuscript deals with an overall important topic demonstrating benefits of a PACU structure. Nevertheless, as several publications dealing with fast-track PACU concepts already exist, the overall new information is low. Nevertheless, the manuscript is overall very well written. However, there exist some issues that may be addressed within a revision:
1) Please elaborate on the rationale for the given PACU indications as most of them in our tertiary care hospital would simple be transferred to a recovery room with discharge time < 2 hours to the normal ward. Is it just the operation and/or also patient´s individual risk stratification, e.g. cardiovascular, pulmonary limiting comorbidities?
2) This is also very important as the authors state that admission to IMCU or PACU was at the discretion of the attending anaesthesiologist. I would rather guess that it should be perioperative team decision based on standard operating procedures (SOP)? Thus, how can the authors be sure that there has been no change in behavior during these resource limiting times?
3) Moreover, even to run a PACU staff is needed, and as far as I am concerned this is mostly the more difficult problem for “missing” beds. This holds even more true as the patient-to-nurse-ratio is lower in the here presented new PACU structure. How did the authors solve this issue at their hospital.
4) Why did you not plan with an anaesthesiologist always in the PACU as this would be the most important quality indicator? In addition, was there a change in delivering of medical treatment in the PACU, e.g. increased short-term catecholaminergic therapy, more (non-)invasive ventilation, mobilization, or was it just a different “monitoring” location compared to IMCU? This is important as the authors did not present a difference in “organ” outcomes. A table thus demonstrating the differences between ICU, IMCU and PACU in terms of staffing and medical treatment options to visualize at once would be of benefit. I understood: PACU was run by the Anesthesiology and IMCU by Surgeons?
5) Were surgeons integrated in the PACU concept?
6) Concerning the primary outcome: at least in my country there has been a huge economic change in terms of revenues (e.g. change in DRG system). This triggered a lot of discharge practices independent of changes in infrastructure/procedural change. This has also been enhanced by the COVID pandemic lowering resources. How you the authors exclude such effects in their analysis? Moreover, is the presented difference in total LOS clinically relevant? I would suggest to only calculate the difference between both groups. Finally, the order to treat patients in the PACU < 24 hours could have also influenced the shorter durations compared to a more liberal discharge strategy on the IMCU. Could you please elaborate on these thoughts?
7) Is the difference in ASA-PS regarded to be clinically relevant? The same holds true for AFib, IDDM. I would thus rather suggest omitting all p-values in the demographic comparison as it is now used in most high impact factor journal. At a first glance, both groups seem to be very homogenous. The same holds true for surgical procedures.
8) Do the authors have any idea about the general costs of IMCU and PACU at their hospital?
Taken together, I would hypothesize that overall there is no clinically relevant difference between IMCU and PACU. I would thus recommend to attenuate the authors´ conclusion.
Reviewer 2 Report
Comments and Suggestions for Authors
This paper presents single institution effects of transition of post-anaesthesia care after noncardiac surgery from Intermediate Care Units (IMCU) to Post-Anaesthesia Care Units (PACU). Despite reasonable methodology, and smooth language, important limitations should be addressed in more details.
1. The difference in hospital length of stay (LOS) in favor of PACU period is statistically significant but clinical significance of 0,4 days (difference in medians between PACU period and ICMU period) is questionable. The lack of difference in postoperative complications is in line with the latter.
2. The paper presents results from single center (teaching hospital in the Netherlands), which raises the concern of generalizability for other hospitals and healthcare systems. This is signaled in the discussion but not sufficiently addressed. In many healthcare systems IMCUs do not exist or they are devoted only for patients previously hospitalized in intensive care units (ICU), and in many centers the real problem is limited number of beds in PACU or lack of PACU (only ICUs and regular surgical wards exist). This limitation makes the paper not relevant for many hospitals and healthcare systems.
Additional comments:
1. Table 1. Is really ASA-PS significantly different despite the same median and IQR?
2. It would be interesting to assess economic impact of PACU (as compared to post-anesthesia/postoperative care in IMCU)?
Round 2
Reviewer 1 Report
Comments and Suggestions for Authors
Dear authors, thank you very much for presenting me the revised and significantly improved version of the manuscript. Congrats!